# Discrete Neural Algorithmic Reasoning

**Gleb Rodionov** [1]   **Liudmila Prokhorenkova** [2]

## Abstract

Neural algorithmic reasoning aims to capture computations with neural networks by training models to imitate the execution of classical algorithms. While common architectures are expressive enough to contain the correct model in the weight space, current neural reasoners struggle to generalize well on out-of-distribution data. On the other hand, classical computations are not affected by distributional shifts as they can be described as transitions between discrete computational states. In this work, we propose to force neural reasoners to maintain the execution trajectory as a combination of finite predefined states. To achieve this, we separate discrete and continuous data flows and describe the interaction between them. Trained with supervision on the algorithm's state transitions, such models are able to perfectly align with the original algorithm. To show this, we evaluate our approach on multiple algorithmic problems and achieve perfect test scores both in single-task and multitask setups. Moreover, the proposed architectural choice allows us to prove the correctness of the learned algorithms for any test data.

## 1. Introduction

Learning to capture algorithmic dependencies in data and to perform algorithmic-like computations with neural networks is a core problem in machine learning, long studied using various approaches (Khardon & Roth, 1997; Graves et al., 2014; Zaremba & Sutskever, 2014; Reed & De Freitas, 2015; Kaiser & Sutskever, 2015; Veličković et al., 2020b).

Neural algorithmic reasoning (Veličković & Blundell, 2021) is a research area focusing on building models capable of executing classical algorithms. Relying on strong theoreti-

cal guarantees of algorithms to work correctly on any input of any size and distribution, this setting provides unlimited challenges for out-of-distribution generalization of neural networks. Prior work explored this setup using the CLRS-30 benchmark (Veličković et al., 2022), which covers classical algorithms from the Introduction to Algorithms textbook (Cormen et al., 2009) and uses graphs as a universal tool to encode data of various types. Importantly, CLRS-30 also provides the decomposition of classical algorithms into subroutines and simple transitions between consecutive execution steps, called hints, which can be used during training in various forms.

The core idea of the CLRS-30 benchmark is to understand how neural reasoners generalize well beyond the training distribution, namely on larger graphs. Classical algorithms possess strong generalization due to the guarantee that correct execution steps never encounter 'out-of-distribution' states, as all state transitions are predefined by the algorithm. In contrast, when encountering inputs from distributions that significantly differ from the training data, neural networks are usually not capable of robustly maintaining internal calculations in the desired domain. Consequently, due to the complexity and diversity of all possible data that neural reasoners can be tested on, the generalization performance of such models may vary depending on particular test distributions (Mahdavi et al., 2023). Given that, it is becoming important to interpret internal computations of neural reasoners to find errors or to prove the correctness of the learned algorithms (Georgiev et al., 2022).

Interpretation methods have been actively developing recently due to various real-world applications of neural networks and the need to debug and maintain systems based on them. Especially, the Transformer architecture (Vaswani et al., 2017) demonstrates state-of-the-art performance in natural language processing and other modalities, representing a field for the development of interpretability methods (Elhage et al., 2021; Weiss et al., 2021; Zhou et al., 2024; Lindner et al., 2023). Based on active research on a computational model behind the transformer architecture, recent works propose a way to learn models that are fully interpretable by design (Friedman et al., 2023).

We found the ability to design models that are interpretable in a simple and formalized way to be crucial for neural

[1]Yandex Research, Moscow, Russia [2]Yandex Research, Amsterdam, The Netherlands. Correspondence to: Gleb Rodionov <rodionovgleb@yandex-team.ru>.

*Proceedings of the 42nd International Conference on Machine Learning*, Vancouver, Canada. PMLR 267, 2025. Copyright 2025 by the author(s).

algorithmic reasoning as it is naturally related to the goal of learning to perform computations with neural networks.

In this paper, we propose to force neural reasoners to follow the execution trajectory as a combination of finite predefined states, which is important for both generalization ability and interpretability of neural reasoners. To achieve that, we start with an attention-based neural network and describe three building blocks to enhance its generalization abilities: feature discretization, hard attention, and the separation of discrete and continuous data flows. In short, all mentioned blocks are connected:

- State discretization does not allow the model to use complex and redundant dependencies in data;

- Hard attention is needed to ensure that attention weights will not be annealed for larger graphs and to limit the set of possible messages that each node can receive;

- Separating discrete and continuous flows is needed to ensure that state discretization does not lose information about continuous data.

Then, we build fully discrete neural reasoners for different algorithmic tasks and demonstrate their ability to perfectly mimic ground-truth algorithm execution. As a result, we achieve perfect test scores on multiple algorithmic tasks with guarantees of correctness on any test data. Moreover, we demonstrate that a single network is capable of executing all covered algorithms in a multitask setting, also achieving perfect generalization.

In summary, we consider the proposed blocks crucial components for robust and interpretable neural reasoners and demonstrate that discretized models, trained with hint supervision, perfectly capture the dynamics of the underlying algorithms and do not suffer from distributional shifts.

## 2. Background

### 2.1. Algorithmic Reasoning

Performing algorithmic-like computations usually requires the execution of sequential steps and the number of such steps depends on the input size. To imitate such computations, neural networks are expected to be based on some form of recurrent unit, which can be applied to a particular problem instance several times (Zaremba & Sutskever, 2014; Kaiser & Sutskever, 2015; Vinyals et al., 2015; Veličković et al., 2020b).

The CLRS Algorithmic Reasoning Benchmark (CLRS-30) (Veličković et al., 2022) defines a general paradigm of algorithmic modeling based on Graph Neural Networks (GNNs), as graphs can naturally represent different input types and

manipulations over such inputs. Also, GNNs are proven to be well-suited for neural execution (Xu et al., 2020; Dudzik & Veličković, 2022).

The CLRS-30 benchmark covers different algorithms over various domains (arrays, strings, graphs) and formulates them as algorithms over graphs. Additionally, CLRS-30 proposes utilizing the decomposition of the algorithmic trajectory execution into simple logical steps, called hints. Using this decomposition is expected to better align the model to desired computations and prevent it from utilizing hidden non-generalizable dependencies of a particular train set. Prior work demonstrates a wide variety of additional inductive biases for models towards generalizing computations, including different forms of hint usage (Veličković et al., 2022; Bevilacqua et al., 2023), biases from standard data structures (Jain et al., 2023; Jürß et al., 2024), knowledge transfer and multitasking (Xhonneux et al., 2021; Ibarz et al., 2022; Numeroso et al., 2023). Also, recent studies demonstrate several benefits of learning neural reasoners end-to-end without any hints at all (Mahdavi et al., 2023; Rodionov & Prokhorenkova, 2023).

The recently proposed SALSA-CLRS benchmark (Minder et al., 2023) enables a more thorough out-of-distribution (OOD) evaluation compared to CLRS-30 with increased test sizes (up to 100-fold train-to-test scaling, compared to 4-fold for CLRS-30) and diverse test distributions.

Despite significant gains in the performance of neural reasoners in recent work, current models still struggle to generalize to OOD test data (Mahdavi et al., 2023; Georgiev et al., 2023; Minder et al., 2023). While de Luca & Fountoulakis (2024) prove by construction the ability of the transformer-based neural reasoners to perfectly simulate graph algorithms (with minor limitations arising from the finite precision), it is still unclear if generalizable and interpretable models can be obtained via learning. Importantly, the issues of OOD generalization are induced not only by the challenges of capturing the algorithmic dependencies in the data but also by the need to carefully operate with continuous inputs. For example, investigating the simplest scenario of learning to emulate the addition of real numbers, Klindt (2023) demonstrates the failure of some models to exactly imitate the desired computations due to the nature of gradient-based optimization. This limitation can significantly affect the performance of neural reasoners on adversarial examples and larger input instances when small errors can be accumulated.

### 2.2. Transformer Interpretability and Computation Model

Transformer (Vaswani et al., 2017) is a neural network architecture for processing sequential data. The input to the transformer is a sequence of tokens from a discrete vocabu-

lary. The input layer maps each token to a high-dimensional embedding and aggregates it with the positional encoding. The key components of each layer are attention blocks and MLP with residual connections. Providing a detailed description of mechanisms learned by transformer models (Elhage et al., 2021) is of great interest due to their widespread applications.

RASP (Weiss et al., 2021) is a programming language proposed as a high-level formalization of the computational model behind transformers. The main primitives of RASP are elementwise sequence functions, *select* and *aggregate* operations, which conceptually relate to computations performed by different blocks of the model. Later, Lindner et al. (2023) presented Tracr, a compiler for converting RASP programs to the weights of the transformer model, which can be useful for evaluating interpretability methods.

While RASP might have limited expressibility, it supports arbitrarily complex continuous functions that in theory can be represented by the transformer architecture, but are difficult to learn. Also, RASP is designed to formalize computations over sequences of fixed length. Motivated by that, Zhou et al. (2024) proposed RASP-L, a restricted version of RASP, which aims to formalize the computations that are easy to learn with transformers in a size-generalized way. The authors also conjecture that the length-generalization of transformers on algorithmic problems is related to the 'simplicity' of solving these problems in the RASP-L language.

Another recent work (Friedman et al., 2023) describes Transformer Programs: constrained transformers that can be trained using gradient-based optimization and then automatically converted into a discrete, human-readable program. Built on RASP, transformer programs are not designed to be size-invariant.

# 3. Discrete Neural Algorithmic Reasoning

## 3.1. Encode-Process-Decode Paradigm

Our work follows the encode-process-decode paradigm (Hamrick et al., 2018), which is usually employed for step-by-step neural execution.

All input data is represented as a graph $G$ with an adjacency matrix $A$ and node and edge features that are first mapped with a simple linear encoder to high-dimensional vectors of size $h$. Let us denote node features at a time step $t$ ($1 \leq t \leq T$) as $X^t = (x_1^t, \ldots, x_n^t)$ and edge features as $E_t = (e_1^t, \ldots, e_m^t)$. Then, the processor, usually a single-layer GNN, recurrently updates these features, producing node and edge features for the next step:

$$X^{t+1}, E^{t+1} = \text{Processor}(X^t, E^t, A).$$

The processor network can operate on the original graph

defined by the task (for graph problems) or on the fully connected graph. For the latter option, the information about the original graph can be encoded into the edge features.

The number of processor steps $T$ can be defined automatically by the processor or externally (e.g., as the number of steps of the original algorithm). After the last step, the node and edge features are mapped with another linear layer, called the decoder, to the output predictions of the model.

If the model is trained with hint supervision, the changes of node and edge features at each step are expected to be related to the original algorithm execution. In this sense, the processor network aims to mimic the algorithm's execution in the latent space.

## 3.2. Discrete Neural Algorithmic Reasoners

In this section, we describe the constraints for the processor that allow us to achieve a fully interpretable neural reasoner. We start with Transformer Convolution (Shi et al., 2020) with a single attention head.

As mentioned above, at each computation step $t$ ($1 \leq t \leq T$), the processor takes the high-dimensional embedding vectors for node and edge features as inputs and then outputs the representations for the next execution step.

Each node feature vector $x_i$ is projected into *query* ($Q_i$), *key* ($K_i$), and *value* ($V_i$) vectors via learnable parameter matrices $W_Q$, $W_K$, and $W_V$, respectively. Edge features $e_{ij}$ are projected into a *key* ($K_{ij}$) vector with a matrix $W_K^E$. Then, for each directed edge from a node $j$ to a node $i$ in the graph $G$, we compute the attention coefficient

$$\alpha_{ij} = \frac{\langle Q_j, K_i + K_{ij} \rangle}{\sqrt{h}},$$

where $\langle a, b \rangle$ denotes the dot product. Then, each node $i$ normalizes all attention coefficients across its neighbors using the softmax function with temperature $\tau$ and receives the aggregated message:

$$\hat{\alpha}_{ij} = \frac{\exp(\alpha_{ij}/\tau)}{\sum_{k \in N(i)} \exp(\alpha_{ik}/\tau)}, \quad M_i = \sum_{k \in \mathcal{N}(i)} \hat{\alpha}_{ik} V_k, \quad (1)$$

where $\mathcal{N}(i)$ denotes the set of all incoming neighbors of the node $i$ and $M_i$ is the message sent to the $i$-th node.

For undirected graphs, we consider two separate edges in each direction. Also, for each node, we consider a self-loop connecting the node to itself. For multi-head attention, each head $l$ separately computes the messages $M_i^l$ which are then concatenated.

Similar to Transformer Programs, we enforce attention to be hard attention. We found this property important not only for interpretability but also for size generalization, as

hard attention allows us to overcome the annealing of the attention weights for arbitrarily large graphs and strictly limit the set of messages that each node can receive.

After the message computation, node and edge features are updated depending on the current values and received messages using feed-forward MLP blocks:

$$\hat{x}_i^{t+1} = \text{FFN}_{nodes}([x_i^t, M_i^t]),$$
$$\hat{e}_{ij}^{t+1} = \text{FFN}_{edges}([e_{ij}^t, \hat{\alpha}_{ji}V_i, \hat{\alpha}_{ij}V_j]).$$

We also enforce all node and edge features to be from a fixed finite set, which we call *states*. We ensure this property by adding discrete bottlenecks at the end of the processor block:

$$x_i^{t+1} = \text{Discretize}_{nodes}(\hat{x}_i^{t+1}),$$
$$e_{ij}^{t+1} = \text{Discretize}_{edges}(\hat{e}_{ij}^{t+1}).$$

We implement discretization by projecting the features into vectors of size $k$ (states logits) which we then use to produce a one-hot vector corresponding to a discrete state. During training with hints, we use teacher forcing and directly use ground-truth states as inputs for the next processor step. For no-hint learning, we apply Gumbel-Softmax (Jang et al., 2017) with annealing temperature to the states logits. During inference, we simply use the argmax function over the states logits dimension.

### 3.3. Continuous Inputs

Clearly, most algorithmic problems operate with continuous or unbounded inputs (e.g., weights on edges). Usually, all input data is encoded into node and edge features and the processor operates over the resulting vectors. The proposed discretization of such vectors would lead to the loss of information necessary for performing correct execution steps. To handle such inputs (which we will call *scalars* referring to both continuous and size-dependent integer inputs such as node indexes) one option is to use Neural Execution Engines (Yan et al., 2020) that enable operating with bit-wise representations of integer and (in theory) real numbers. Such representations are bounded by design, but fully discrete and interpretable.

We propose another option: to maintain scalar inputs (denote them by $S$) separately from the node and edge features and use them only as edge priorities $s_{ij}$ in the attention block. If scalars are related to the nodes, we assign them to edges depending on the scalar of the sender or receiver node. We implement this simply by augmenting the key vectors $K_{ij}$ of each edge with the indicator of whether the given edge has the "best" (min or max) scalar among the other edges to node $j$ with the same discrete state. Thus, scalars affect only the attention weights, not the messages or the node states.

For multiple different scalar inputs (e.g., weighted edges

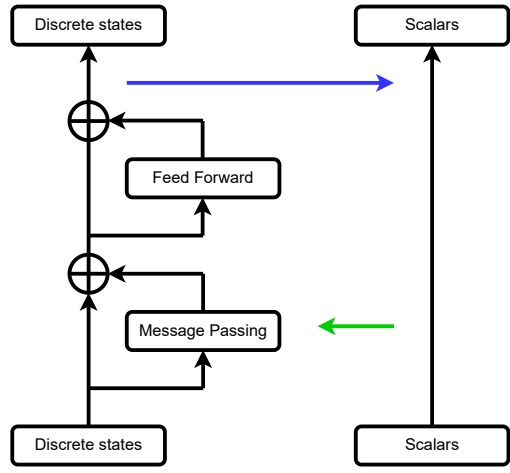

*Figure 1.* An illustration of the proposed separation between discrete and continuous data flows. Scalars can only affect the attention weights (Green) and can be modified with actions via ScalarUpdate (Blue).

and node indexes), we use multi-head attention, where each head operates with separate scalars.

Thus, the interface of the proposed processor can be described as

$$X^{t+1}, E^{t+1} = \text{Processor}(X^t, E^t, A, S),$$

where $X^t, X^{t+1}$ and $E^t, E^{t+1}$ are from fixed sets. State sets are independent of the execution step $t$ and the input graph (including scalar inputs $S$).

### 3.4. Manipulations over Continuous Inputs

The proposed selector offers a read-only interface to scalar inputs, which is not expressive enough for most algorithms. However, we note that the algorithms can be described as *discrete* manipulations over input data. For example, the Dijkstra algorithm (Dijkstra, 1959) takes edge weights as inputs and uses them to find the shortest path distances. Computed distances can affect the subsequent execution steps. We note that such distances can be described as the sum of the weights of the edges that form the shortest path to the given node. In other words, the produced scalars depend only on input scalars and discrete execution states.

To avoid the challenges of learning continuous updates with high precision (Klindt, 2023), we propose to learn discrete manipulations with scalars. The updated scalars can then be used with the described selector in the next steps.

In our experiments, we use a scalar updater capable of incrementing, moving, and adding scalars depending on discrete

node/edge states:[1]

$$s_i^{t+1} = \text{inc}(x_i^t) + \text{keep}(x_i^t) \cdot s_i^t + \sum_{j \in \mathcal{N}(i)} \text{push}(e_{ji}^t) \cdot s_{ji}^t,$$

$$s_{ij}^{t+1} = \text{inc}(e_{ij}^t) + \text{keep}(e_{ij}^t) \cdot s_{ij}^t + \text{push}(x_i^t) \cdot s_i^t, \quad (2)$$

where $s_i$ are node-related scalars, $s_{ij}$ are edge-related scalars, and $\text{inc}$, $\text{keep}$, $\text{push}$ are 0-1 functions representing whether a scalar in each node/edge should be incremented, kept, or pushed to any of its neighbors. We implement these functions as simple linear projections of node/edge features with subsequent discretization.

Finally, our proposed method (see Figure 1) can be described as:

$$X^{t+1}, E^{t+1} = \text{Processor}(X^t, E^t, A, S^t),$$

$$S^{t+1} = \text{ScalarUpdate}(X^t, E^t, A, S^t).$$

The proposed neural reasoners are fully discrete and can be interpreted by design. Moreover, the proposed selector block guarantees predictable behavior of the message passing across all graph sizes, as it compares discrete state importance and uses continuous scalars only to break ties between equally important states.

# 4. Experiments

In this section, we perform experiments to evaluate how the proposed discretization affects the performance of neural reasoners on diverse algorithmic reasoning tasks. Our main questions are:

1. Can the proposed discrete neural reasoners capture the desired algorithmic dynamics with hint supervision?

2. How does discretization affect OOD and size-generalization performance of neural reasoners?

3. Is the proposed model capable of multi-task learning?

Additionally, we are interested in whether discrete neural reasoners can be learned without hints and how they tend to utilize the given number of node and edge states to solve the problem. We discuss no-hint experiments separately in Section 6.

Finally, we investigate whether graph sizes in the train set can be reduced without compromising generalization performance; see Appendix B for details.

---

[1]We additionally investigate if the operation set can be extended, see Appendix C.

## 4.1. Datasets

We perform our experiments on the problems from the recently proposed SALSA-CLRS benchmark (Minder et al., 2023), namely BFS, DFS, Prim, Dijkstra, Maximum Independent Set (MIS), and Eccentricity. We believe that the proposed method is not limited to the covered problems, but we leave the implementation of the required data flows (e.g., edge-based reasoning (Ibarz et al., 2022), graph-level hints, interactions between different scalars) for future work.

The train dataset of SALSA-CLRS consists of random graphs with at most 16 nodes sampled from the Erdös-Rényi (ER) distribution with parameter $p$ chosen to be as low as possible such that graphs remain connected with high probability. The test set consists of sparse graphs of sizes from 16 to 1600 nodes.

We slightly modify the hints from the benchmark without conceptual changes (e.g., we have modified the hints for the DFS problem to remove graph-level hints and node discovery/finish times). Our discrete states are defined by the benchmark's non-scalar hints, while our scalars are exactly the hints of the $scalar$ type (see Veličković et al. (2022) for details on hint design).

## 4.2. Baselines and Evaluation

We compare the performance of our proposed discrete model with two baseline sparse models, GIN (Xu et al., 2019) and Pointer Graph Network (Veličković et al., 2020a). We report both node-level and graph-level metrics for the baselines and our model. Graph-level metrics measure the proportion of test graphs for which all node-level outputs are predicted correctly. For example, for the BFS problem, the output of the algorithm is a tree, where each node points to its parent in the BFS exploration tree (and the starting node points to itself). Pointer is a specific hint/output type which forces each node to point to exactly one node.

Additionally, we compare our model with Triplet-GMPNN (Ibarz et al., 2022) and two recent approaches, namely Hint-ReLIC (Bevilacqua et al., 2023) and G-ForgetNet (Bohde et al., 2024), which demonstrate state-of-the-art performance in hint-based neural algorithmic reasoning. However, as these methods are evaluated on the CLRS-30 benchmark and their code is not yet publicly available, we only compare them on the corresponding tasks (BFS, DFS, Dijkstra, Prim) and the CLRS-30 test data, namely ER graphs with $p = 0.5$ of size 64. Note that this test data is denser than that of SALSA-CLRS, meaning shorter roll-outs for the given tasks. Furthermore, only node-level metrics have been reported for these methods.

### 4.3. Model Details

For our experiments, we use the model described in Section 3. We use one attention head for each scalar value. The number of processor steps is set to the length of the ground-truth algorithm trajectory, which is consistent with prior work. We use one architecture (except the task-dependent encoders/decoders), including the $\mathrm{ScalarUpdate}$ module, for all the problems.

We recall that neural reasoners can operate either on the original graph (which is defined by the problem) or on denser graphs with the original graph encoded into edge features. SALSA-CLRS proposes to enhance the size-generalization abilities of neural reasoners with increased test sizes (up to 100-fold train-to-test scaling), so we use the original graph for message-passing, similar to the SALSA-CLRS baselines. We also add a virtual node that communicates with all nodes of the graph.

### 4.4. Training Details and Hyperparameters

We train each model for 1000 optimization steps using the Adam optimizer with a learning rate of $\eta = 0.001$ and a batch size of 32 graphs; teacher forcing is applied throughout training. For the discrete bottlenecks (attention weights and $\mathrm{ScalarUpdate}$ operations), we anneal the softmax temperature geometrically from 3.0 to 0.01, decreasing it at each training step. A complete list of all hyperparameters is provided in our source code.[2]

During training, we minimize the standard hint and output losses: scalar hints are optimized with the MSE loss, and the other types of hints are optimized with the cross-entropy and categorized cross-entropy losses (Veličković et al., 2022; Ibarz et al., 2022). Notably, we do not supervise any additional details in model behavior, such as selecting the most important neighbor in the attention block or the exact operations with scalars.

For multitask experiments, we adopt the setup from Ibarz et al. (2022) and train a single processor with task-dependent encoders/decoders to imitate all covered algorithms simultaneously. We make 10000 optimization steps on the accumulated gradients across each task and keep all hyperparameters the same as in the single task.

Our models are trained on a single A100 GPU, requiring less than 1 hour for single-task and 5-6 hours for multitask training.

### 4.5. Results

We found learning with teacher forcing suitable for discrete neural reasoners, as discretization blocks allow us to perform the exact transitions between the states. Trained with

---

[2] https://github.com/yandex-research/dnar

---

*Table 1.* Node/graph-level test scores for the proposed discrete reasoner and the baselines on SALSA-CLRS test data. Scores are averaged across 5 different seeds; standard deviation is omitted. For the eccentricity problem, only the graph-level metric is applicable.

| TASK | SIZE | GIN | PGN | DNAR (OURS) |
|------|------|-----|-----|-------------|
| BFS | 16 | 98.8 / 92.5 | 100. / 100. | 100. / 100. |
| | 80 | 95.3 / 59.4 | 99.8 / 88.1 | 100. / 100. |
| | 160 | 95.1 / 37.8 | 99.6 / 66.3 | 100. / 100. |
| | 800 | 86.9 / 0.9 | 98.7 / 0.2 | 100. / 100. |
| | 1600 | 86.5 / 0.0 | 98.5 / 0.0 | 100. / 100. |
| DFS | 16 | 41.5 / 0.0 | 82.0 / 19.9 | 100. / 100. |
| | 80 | 30.4 / 0.0 | 38.4 / 0.0 | 100. / 100. |
| | 160 | 20.0 / 0.0 | 26.9 / 0.0 | 100. / 100. |
| | 800 | 19.5 / 0.0 | 24.9 / 0.0 | 100. / 100. |
| | 1600 | 17.8 / 0.0 | 23.1 / 0.0 | 100. / 100. |
| SP | 16 | 95.2 / 49.8 | 99.3 / 89.5 | 100. / 100. |
| | 80 | 62.4 / 0.0 | 94.2 / 3.3 | 100. / 100. |
| | 160 | 53.3 / 0.0 | 92.0 / 0.0 | 100. / 100. |
| | 800 | 40.4 / 0.0 | 87.1 / 0.0 | 100. / 100. |
| | 1600 | 36.9 / 0.0 | 84.5 / 0.0 | 100. / 100. |
| PRIM | 16 | 89.6 / 29.7 | 96.4 / 69.9 | 100. / 100. |
| | 80 | 51.6 / 0.0 | 79.7 / 0.0 | 100. / 100. |
| | 160 | 49.5 / 0.0 | 75.6 / 0.0 | 100. / 100. |
| | 800 | 45.0 / 0.0 | 69.5 / 0.0 | 100. / 100. |
| | 1600 | 43.2 / 0.0 | 66.8 / 0.0 | 100. / 100. |
| MIS | 16 | 79.9 / 3.3 | 99.8 / 98.6 | 100. / 100. |
| | 80 | 79.9 / 20.0 | 99.4 / 88.9 | 100. / 100. |
| | 160 | 78.2 / 0.0 | 99.4 / 76.2 | 100. / 100. |
| | 800 | 83.4 / 0.0 | 98.8 / 18.0 | 100. / 100. |
| | 1600 | 79.2 / 0.0 | 98.9 / 5.2 | 100. / 100. |
| ECC. | 16 | NA / 25.3 | NA / 100. | NA / 100. |
| | 80 | NA / 23.8 | NA / 100. | NA / 100. |
| | 160 | NA / 26.1 | NA / 100. | NA / 100. |
| | 800 | NA / 17.1 | NA / 100. | NA / 100. |
| | 1600 | NA / 16.0 | NA / 83.0 | NA / 100. |

---

step-wise hint supervision, discrete neural reasoners are able to perfectly align with the original algorithm and generalize on larger test data without any performance loss. We report the evaluation results in Tables 1 and 2. Also, our multitask experiments show that the proposed discrete models are capable of multitask learning and demonstrate the perfect generalization scores in a multitask manner too.

Recall that we have three key components of our contribution: feature discretization, hard attention, and separating discrete and continuous data flows. To evaluate the importance of each component for generalization capabilities of the proposed models, we conduct an ablation study; the details can be found in Appendix A. In short, we demonstrate that removing each of these components yields the model without provable guarantees of perfect generalization. However, these components differ in terms of the impact

*Table 2.* Node-level test scores for the proposed discrete reasoner and the baselines on CLRS-30 test data. Test graphs are of size 64. Scores are averaged across 5 different seeds.

| TASK | HINT-RELIC | G-FORGETNET | DNAR (OURS) |
|------|-----------|-------------|-------------|
| BFS | $99.00 \pm 0.2$ | $99.96 \pm 0.0$ | $100. \pm 0.0$ |
| DFS | $100. \pm 0.0$ | $74.31 \pm 5.0$ | $100. \pm 0.0$ |
| SP | $97.74 \pm 0.5$ | $99.14 \pm 0.1$ | $100. \pm 0.0$ |
| PRIM | $87.97 \pm 2.9$ | $95.19 \pm 0.3$ | $100. \pm 0.0$ |

on the performance. In particular, using regular attention instead of hard attention yields perfect test scores for given datasets, but it is possible to construct adversarial examples with large neighborhood sizes where performance drops. On the other hand, removing discretization from the scalar updater significantly affects the performance even on the small test graphs.

## 5. Interpetability and Testing

In addition to the empirical evaluation of the trained models on diverse test data, the discrete and size-independent design of the proposed models allows us to interpret and test them manually. Namely, we can validate that the model will perform the exact discrete state transitions (including discrete operations with scalars) as the original algorithm.

First, due to the hard attention, each node receives exactly one message. Also, the message depends only on the discrete states of the corresponding nodes and edges. Thus, node and edge states after a single processor step depend only on the current states and the received message. Therefore, all the possible options can be directly enumerated to test if all states change to the correct ones.

The final component to fully interpret the whole model is the attention block. We note that our implementation simply augments the key vectors with the indicators of whether the given edge has the best scalar among others. For example, it may happen that for some state, the maximum attention weight is achieved for an edge without the indicator. However, given the finite number of discrete states, we can manually check the unnormalized attention scores between each pair of states both with this indicator and without it. Combined with hard attention, this would imply that the attention block attends to nodes based on their predefined states and uses scalar priorities only to break ties. A particular example with a more detailed explanation for the BFS problem can be found in Appendix D.

Given that, we can unit-test all possible state transitions and attention blocks. With full coverage of such tests, we can guarantee the correct execution of the desired algorithm for *any* test data. We tested our trained models from Section 4

by manually verifying state transitions. As a result, we confirm that the attention block indeed operates as $select\_best$ selector, as the model actually uses these indicators to increase the attention weights. Thus, we can guarantee that for any graph size, the model will mirror the desired algorithm, which is correct for any test size.

## 6. Towards No-Hint Discrete Reasoners

In this section, we discuss the challenges of training discrete reasoners without hints, which can be useful when tackling new algorithmic problems.

Training deep discretized models is known to be challenging (Mahdavi et al., 2023). We observe that without hyperparameter search, discrete models are only slightly improved over the untrained models. Therefore, we focus only on the BFS algorithm, as it is well-aligned with the message-passing framework, has short roll-outs, and can be solved with a small number of states (note that for no-hint models node/edge state count is a hyperparameter due to the absence of the ground-truth states trajectory).

We recall that the output of the BFS problem is the exploration tree pointing from each node to its parent. Each node chooses as a parent the neighbor from the previous distance layer with the smallest index.

We perform hyperparameter search over graph sizes in the train set (using ER graphs with $p = 0.5$ and $n \in [4, 16]$), discrete node state count (from 2 to 6 states), softmax temperature annealing schedules ($[3, 0.01], [3, 0.1], [3, 1]$). For each hyperparameter choice, we train 5 models with different seeds. We validate the resulting models on the graphs of size 16. The best resulting model is obtained with the graph size $n = 5$ and 4 node states. The trained models never achieved perfect validation scores (see Table 3 for the results).

*Table 3.* BFS node/graph-level scores of the *best_no_hint_model* for different graph sizes.

| | 5 | 16 | 64 |
|---|---|----|----|
| *best_no_hint_model* | 99 / 86 | 94 / 34 | 79 / 0 |

Then, we select the best-performing models and try to analyze the errors of the resulting models and reverse-engineer how they utilize the given states. First, we look at the node states after the last step of the processor and note that the states correspond to the distances from the starting node. More formally, we note that the model with four states uses the first state for the starting node, the second state for its neighbors, the third state for nodes at distance two from the starting node, and the last state for all other nodes. Such states-based classification of distance has ac-

curacy $> 98\%$ when tested on 1000 random graphs with 16 nodes. We observe that for the nodes that belong to the first four distance layers from the starting node, the pointers are predicted with 100% accuracy and these pointers are computed layer-by-layer as in the ground-truth algorithm (we refer to Appendix E for illustrations). The model's errors occur at distances $\geq 4$ from the starting node (we did not reverse-engineer the specific logic of computations on larger distances).

We found this behavior well-aligned with the BFS algorithm, which indicates the possibility of achieving the perfect validation score with enough number of states. However, this algorithm does not generalize since it fails at distances larger than those encountered during training.

On the other hand, one can demonstrate that for a small enough state count (for BFS, it is two node states and two edge states) and sufficiently diverse validation data, the perfect validation performance implies that the learned solution will generalize to any graph size.

Therefore, we highlight the need to achieve perfect validation performance with models that use as few states as possible, which corresponds to the minimum description length (MDL) theory (Myung, 2000; Rissanen, 2006) and is related to the notion of Kolmogorov complexity (Kolmogorov, 1963).

Finally, we note that for sequential problems, such as DFS, obtaining a good no-hint model can be even more challenging and may require additional effort. One possible way to overcome this limitation is to implement a curriculum learning setup, which we leave for future work.

## 7. Limitations and Future Work

**Limitations** In this work, we propose a method to learn robust neural reasoners that demonstrate perfect generalization performance and are interpretable by design. In this section, we describe some limitations of our work and important directions for future research.

First, several proposed design choices strictly reduce the expressive power of the model. For example, due to the hard attention, the proposed model is unable to compute the mean value from all the neighbors in a single message-passing step, which is trivial for attention-based models (note that this can be computed in several message-passing steps). Thus, the model in its current form is unable to express transitions between hints for some algorithms in the CLRS-30 benchmark in a single processor step.

However, we believe that the expressivity of the proposed model can be enhanced with additional architectural modifications (e.g., edge-based reasoning (Ibarz et al., 2022), global states, interactions between different scalars) that can

be combined with the proposed discretization ideas. Additionally, there are several potential ways to balance between expressive power of the model and strong generalization guarantees, e.g., removing hard attention or removing feature discretization (while keeping the discretization inside the ScalarUpdate module).

Second, while we report perfect scores for the covered tasks, we cannot guarantee that the training will converge to the correct model for any initialization/training data distributions. However, we empirically found the proposed method to be quite robust to various architecture/training hyperparameter choices.

**Future work** Our method is based on the particular architectural choice and actively utilizes the attention mechanism. However, the graph deep learning field is rich in various architectures exploiting different inductive biases and computation flows. The proposed separation between discrete states and continuous inputs may apply to other models. However, any particular construction may require additional effort.

Also, we provide only one example of the ScalarUpdate block. We believe that utilizing a general architecture (e.g., some form of discrete Neural Turing Machine (Graves et al., 2014; Gulcehre et al., 2016)) capable of executing a wider range of manipulations is of interest for future work.

With the development of neural reasoners and their ability to execute classical algorithms on abstract data, it is becoming more important to investigate how such models can be applicable in real-world scenarios according to the Neural Algorithmic Reasoning blueprint (Veličković & Blundell, 2021) and transfer their knowledge to high-dimensional noisy data with intrinsic algorithmic dependencies. While there are several examples of NAR-based models tackling real-world problems (Beurer-Kellner et al., 2022; Numeroso et al., 2023), there are no established benchmarks for extensive evaluation and comparison of different approaches.

Lastly, we leave for future work a deeper investigation of learning interpretable neural reasoners without hints, which we consider essential both from a theoretical perspective and for practical applications, e.g., combinatorial optimization.

## 8. Conclusion

In this paper, we force neural reasoners to maintain the execution trajectory as a combination of finite predefined states. To achieve that, we separate discrete and continuous data flows and describe the interaction between them. The obtained discrete reasoners are interpretable by design. Moreover, trained with hint supervision, such models perfectly capture the dynamics of the underlying algorithms and do not suffer from distributional shifts. We consider

discretization of hidden representations a crucial component for robust neural reasoners.

## Impact Statement

This paper presents work whose goal is to advance the field of Machine Learning. There are many potential societal consequences of our work, none which we feel must be specifically highlighted here.

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

## A. Ablation Study

Recall that we have three key components of our contribution: feature discretization, hard attention, and separating discrete and continuous data flows. In this section, we study the importance of these components for generalization capabilities of the proposed models.

**Discrete bottlenecks**    First, we evaluate the model without any of its discrete bottlenecks: the result is a simple Transformer Convolution processor, which performs comparably to the other baseline models (Table 4).

*Table 4.* Node/graph-level test scores for our base model without all discrete bottlenecks. Scores are averaged across 5 different seeds; standard deviation is omitted.

| SIZE | 16 | 80 | 160 | 800 | 1600 |
|------|------|------|------|------|------|
| BFS | 99.9 / 99.3 | 99.7 / 88.2 | 99.5 / 57.9 | 98.4 / 0.0 | 97.2 / 0.0 |
| DFS | 79.2 / 6.8 | 41.1 / 0.0 | 28.1 / 0.0 | 24.7 / 0.0 | 21.9 / 0.0 |
| SP | 99.3 / 88.7 | 94.1 / 12.4 | 90.3 / 0.0 | 86.9 / 0.0 | 82.4 / 0.0 |
| PRIM | 95.1 / 72.7 | 82.6 / 0.0 | 79.7 / 0.0 | 68.1 / 0.0 | 66.0 / 0.0 |
| MIS | 99.8 / 98.6 | 99.6 / 86.1 | 99.2 / 69.0 | 97.1 / 11.9 | 96.3 / 0.0 |
| ECC. | NA / 79.2 | NA / 41.1 NA / | NA / 28.1 | NA / 24.7 | NA / 21.9 |

**Hard attention**    To highlight the importance of the hard attention for strong size generalization, we train the proposed model but with the regular attention mechanism on the BFS task. The resulting model also demonstrates perfect scores for the given test data. However, the standard test data (the Erdös-Renyi graphs with low edge probability) does not contain nodes with a large number of neighbors, while such nodes can be problematic due to the annealing of the attention weights. Thus, we additionally test the resulting model on the complete bipartite graphs $K_{2,n-2}$ for different $n$. For each $n$, we assign the starting node from the smaller component and test if the second node in this component correctly selects its parent in the BFS tree (Figure 2).

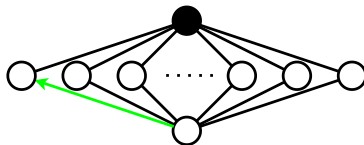

*Figure 2.* Complete bipartite graphs $K_{2,n-2}$ used to evaluate the effect of the attention weight annealing. The black node is the starting node. The highlighted edge (Green) is the ground-truth pointer from the bottom node to its parent in the BFS tree.

Our experiments demonstrate that the model without hard attention fails to predict the correct pointer for larger graphs due to the attention weight annealing (see Table 5). We note that the models from Section 4 are provably correct on any test data.

*Table 5.* Attention weights for the ground-truth pointer (green pointer from Figure 2) for different graph sizes; (+/-) denotes whether the correct pointer was predicted.

| SIZE | 16 | 80 | 160 | 800 | 1600 |
|------|------|------|------|------|------|
| ATTENTION WEIGHT | 0.97 (+) | 0.86 (+) | 0.76 (+) | 0.38 (-) | 0.24 (-) |

**Scalar updater**    As demonstrated in Klindt (2023), simple neural networks trained to sum two real numbers fail to learn the structure of the task and struggle to extrapolate beyond the training data distributions. In this section, we study how these limitations affect the overall performance of neural reasoners to highlight the importance of the proposed discrete manipulations with scalars.

First, we recall that usually all input data is encoded into node and edge features and the processor operates over the resulting vectors. Then, hints of type scalar are directly predicted from the node/edge features. To evaluate the effect of non-discrete ScalarUpdate modules, we simply replace the proposed discrete ScalarUpdate module with a single-layer transformer convolution network, which inputs scalars and node/edge states and outputs the scalars for the next step, keeping the remaining architecture the same as in the main experiments. We train the resulting model on Dijkstra and MST problems.

Additionally, we evaluate the non-discrete ScalarUpdate module in a more straightforward setup. Similarly to Klindt (2023), we train a 2-layer MLP to add two real numbers and use the resulting model as a ScalarUpdate module for the

Dijkstra algorithm. We manually use the learned addition module when the node distances are updated (e.g., the distance of the node $u$ is updated with the sum of the distance of $v$ and the edge $(v, u)$ cost), and use the ground-truth scalars for other scalar updates. Our experiments demonstrate that the resulting model outperforms the baselines on the test size of 16 nodes, but does not generalize well on larger graphs (see the evaluation results in Table 6).

*Table 6.* Node/graph-level test scores for the proposed model with ScalarUpdate replaced by a regular attention-based network trained to predict hints of type scalar. 'Addition only' means that ScalarUpdate is replaced by a 2-layer MLP trained to predict the sum of two numbers (the other values are taken from the ground truth).

| SIZE | 16 | 80 | 160 | 800 | 1600 |
|------|-----|-----|-----|-----|------|
| DIJKSTRA | 99.3 / 94.6 | 60.7 / 0.0 | 42.8 / 0.0 | 19.0 / 0.0 | 11.8 / 0.0 |
| MST | 99.8 / 98.1 | 98.2 / 54.1 | 97.2 / 28.1 | 95.5 / 0.0 | 91.73 / 0.0 |
| DIJKSTRA (ADDITION ONLY) | 99.8 / 96.6 | 95.3 / 71.0 | 86.5 / 46.3 | 41.6 / 3.1 | 22.2 / 0.0 |

We note that all the resulting models demonstrate perfect scores when evaluated with teacher-forced ground-truth scalars. Thus, all state transitions are learned correctly and imperfect test scores are fully described by the errors in manipulations with continuous values.

To summarize, small errors in manipulations with scalars (even restricted to the simplest addition sub-task) severely affect the overall performance of the model, highlighting the importance of the proposed discrete manipulations with scalars.

## B. Minimal Training Sizes

In this section, we highlight an important property of the proposed models: the correct state transitions can be learned even from a trivially small size.

For example, for the BFS problem, it is enough to use graphs with only 3 nodes to observe whether an unvisited node becomes visited depending on the received message. However, the subtask of selecting the parent from multiple visited neighbors requires at least 4 nodes (where the minimum sufficient example is the complete bipartite graph $K(2, 2)$).

To demonstrate that, we conducted additional experiments to empirically find the smallest graph size for perfect fitting of each covered algorithm. For this experiment, we used training with hints: we trained our models for each problem on $\mathrm{ER}(n, 0.5)$ graphs for different $n$ and tested the resulting models on graphs with 160 nodes. Table 7 presents the results. Note that the empirical bound is around 4-5 nodes.

*Table 7.* Node-level test scores for the proposed discrete reasoner on test graphs with 160 nodes, across different training sizes.

| TASK | 3 nodes | 4 nodes | 5 nodes |
|------|---------|---------|---------|
| BFS | 41 | 100 | 100 |
| DFS | 38 | 100 | 100 |
| DIJKSTRA | 13 | 26 | 100 |
| MST | 11 | 14 | 100 |
| MIS | 79 | 100 | 100 |
| ECC. | 45 | 100 | 100 |

## C. ScalarUpdate **Module**

In this section, we first provide an example demonstrating the role of the proposed ScalarUpdate module and then investigate whether this module can be successfully extended to support more complex manipulations with scalars.

To give an example, consider the Dijkstra algorithm. The algorithm uses edge weights to compute the shortest distance from the starting node to each node, building a tree step-by-step. When adding a new node $B$ to the tree and assigning a node $A$ as a parent in the tree, the algorithm computes the distance from the starting node the node $B$ by summing the distance to the node A with the weight of the edge $AB$. To mimic this behavior with our model, we use the ScalarUpdate module. At node level, this module receives as inputs the discrete states of the node and its neighbors and the scalar values stored in the corresponding edges. Depending on the discrete states, this module updates the scalar values by one of the supported operations, as shown in the formula (2). In our example, this module would push the scalar (the distance from the starting node to $A$) from the node $A$ to the edge $AB$ and sum it with the edge's weight. Thus, for edges in the shortest path tree, the scalar value on edges represents the exact shortest distance from the starting node, and for other edges the scalar value will represent the edge weight.

This module allows one to perform any (predefined) operation with scalars needed for the algorithms, while guaranteeing that the correct operation is chosen based on the discrete states, thus being robust to OOD scalar values.

We note that simple manipulations with scalars cover a significant part of the classical algorithms. In this work, we use the minimum set of the required functions, but this set can be directly extended with other functions. Importantly, as ScalarUpdate can be viewed as a separate module, we can separately check if it is possible to train it with any given set of predefined manipulations for any problem only with supervision on the results.

Let us formalize the problem: each input to the ScalarUpdate module can be described as an object with a discrete state $s_i$ (from a fixed predefined set) and several scalar values (we consider two scalars $x_i$ and $y_i$). Note that we omit the separation between nodes and edges and consider objects with several scalar values. For each discrete state $s_i$, there exists a ground-truth update of scalars, e.g., $f(s, x, y) = x + \cos(y)$. The output of the scalar updater can be viewed as a sum:

$$\text{ScalarUpdate}(s, x, y) = \sum_{g \in \mathcal{OPS}} \text{g}(x, y) \cdot \text{activate}_\text{g}(s),$$

where $\mathcal{OPS}$ is a predefined set of operations and $\text{activate}_\text{g}$ is a 0-1 function representing whether a specific operation should be applied. Note that $\text{activate}_\text{g}$ depends only on the discrete state of the input.

For our additional experiment, we train several different ScalarUpdate modules with an extended set of operations:

- $g_0(x, y) = 1$;
- $g_1(x, y) = x$;
- $g_2(x, y) = \cos(x)$;
- $g_3(x, y) = x \cdot y$;
- $g_4(x, y) = \text{atan2}(x, y)$

to learn the following set of ground-truth updates simultaneously:

- $f_0(x, y) = x$;
- $f_1(x, y) = \cos(x)$;
- $f_2(x, y) = \cos(x) + x \cdot y$;
- $f_3(x, y) = \text{atan2}(x, y)$;
- $f_4(x, y) = 1 + x + \text{atan2}(x, y)$.

In particular, we consider a set of 16 discrete states (numbered from 0 to 15 and sampled uniformly) and the ground-truth scalar update is derived from these states by taking the remainder of the division by 5 (updates count).

The learnable parameters of the ScalarUpdate are state embeddings and linear projections for each indicator. We train ScalarUpdate to minimize the MSE loss between the ground-truth and predicted outputs with 5000 optimization steps. Additionally, we train a non-discrete scalar updater (2-layer MLP), similar to our ablation experiments. We refer to the source code for the experiment details.

Inspired by Klindt (2023), we generate training scalars $X$ and $Y$ from $\text{Uniform}[0.5, 1.0]$ and generate the test set by sampling scalars from the $\text{Uniform}[0., 0.5]$ distribution.

We report the evaluation results in Table 8. The proposed discrete ScalarUpdate module successfully learned the correct operations for updates $f_1, ..., f_4$ for all seeds and for $f_0$ in 3 out of 5 runs (note that the model was trained to predict different manipulations for different states simultaneously). For unsuccessful runs, when $f_0$ was not learned correctly, the learned operation for $f_0$ was $g_3$ for some states (i.e., $x \cdot y$ instead of $x$), which can be explained by optimization challenges as the distribution of $y$ is close to 1.

Our experiment demonstrates that the proposed ScalarUpdate module can be extended to support a wider range of manipulations with scalars. We note that such an extension might complicate the optimization problem of selecting the correct operations/operands from the operations results (e.g., such decomposition might not be unique).

*Table 8.* MSE for train/test distributions for the discrete and non-discrete ScalarUpdate modules and different operations.

| | $f_0$ | $f_1$ | $f_2$ | $f_3$ | $f_4$ |
|---|---|---|---|---|---|
| discrete | 0.01 / 0.1 | 0. / 0. | 0 / 0 | 0 / 0 | 0 / 0 |
| non-discrete | 5.7$e$-6 / 0.03 | 5.1$e$-6 / 0.001 | 1.1$e$-5 / 0.007 | 1.0$e$-5 / 0.08 | 2.0$e$-5 / 0.03 |

## D. Interpetability and Testing Details

In this section, we provide additional details on interpretability and testing of the proposed discrete reasoners.

As an example, consider the BFS algorithm. First, recall the pseudocode of the algorithm:

$Starting\_node \leftarrow visited$
$All\_other\_nodes \leftarrow not\_visited$
**for** step **in range(T) do**
  **for** node $U$ **in a graph do**
    **if** $U$ is visited on previous steps **then**
      `continue`
    **end if**
    **if** $U$ has a neighbor $P$ that visited on previous steps **then**
      $U \leftarrow visited$ on this step
      $U$ selects the smallest-indexed such neighbor $P$ as parent:
      Edge $(U, P) \leftarrow pointer$
      Self-loop $(U, U) \leftarrow not\_pointer$
    **end if**
  **end for**
**end for**
**return** a BFS tree described by pointers

Now let us describe how we can verify that the trained DNAR model will perfectly imitate this algorithm for any test data.

First, we note that for each node $U$, the node state on the step $t + 1$ (denoted by $U_{t+1}$) is the function of $U_t$ and $V_t$, where $V$ is the node that sends the message to $U$ on step $t$:

$$U_{t+1} = \text{StateUpdate}(U_t, message\_from\_V_t)$$

How does the node $U$ select a node that will send a message to it? For any node $V$ connected to $U$, the node $U$ computes attention scores depending on discrete states of $V$ of each node and a discrete indicator of whether each node has the smallest (or largest) scalar among all neighbors of $U$ with the same discrete state as $V$. Then, the node $U$ selects the node $V$ with the largest attention score.

In our case, the attention scores only depend on the tuples ($U_{state}$, $V_{state}$, $indicator\_if\_u\_has\_the\_smallest\_index$) and there are only 8 such tuples. We can directly compute these attention scores and verify the required invariants, e.g.,

$$\text{Attention}(not\_visited, visited, smallest) > \text{Attention}(not\_visited, *any\_other*),$$

which would imply that the not_visited node will receive the message from the smallest-indexed visited neighbor if such exists independently of the graph size and distribution. If there is no such neighbor, the node $U$ will receive the message from another not_visited node (or from itself).

After verifying the correctness of the message flows, we need to ensure that the state updates are computed correctly, e.g.,

$$visited = \text{StateUpdate}(not\_visited, message\_from\_visited),$$
$$visited = \text{StateUpdate}(visited, *any*),$$
$$not\_visited = \text{StateUpdate}(not\_visited, message\_from\_not\_visited).$$

The main idea is that due to the finite state count and discrete manipulations with scalars, there are only finite amounts of such checks that can cover all possible state transitions and all of them should be evaluated only once.

# E. State Usage for No-Hint Models

In this section, we illustrate the internal states and pointer prediction dynamics of a model trained for BFS without hint supervision (Figures 3-5). Our analysis suggests that no-hint models with $K$ states tend to use these states as distances from the starting node, with the distances $\geq K$ merged into the same state. Also, pointer predictions for the first $K$ BFS layers are correct and computed layer-by-layer as in the ground-truth algorithm. Errors occur at layers beyond $K$, where some pointers are predicted earlier than in the ground-truth algorithm (Figure 5c). For clarity, we use the model with $K = 3$ discrete states.

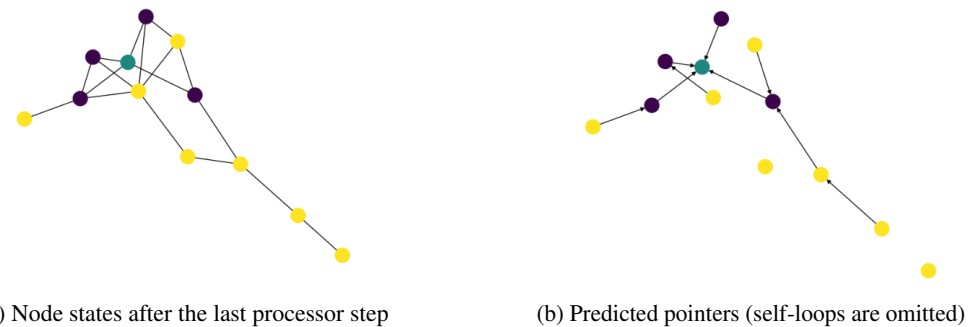

(a) Node states after the last processor step       (b) Predicted pointers (self-loops are omitted)

*Figure 3.* Node states and predicted pointers after the last processor step of the DNAR model (with 3 states) trained without hints. Node colors correspond to their states and the green node is the starting node for BFS.

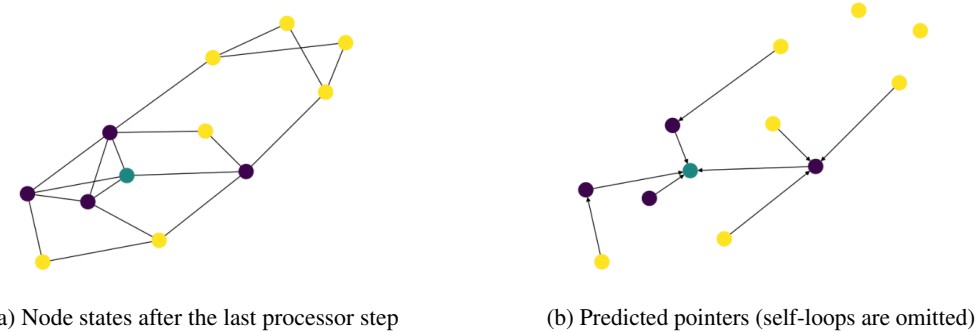

(a) Node states after the last processor step       (b) Predicted pointers (self-loops are omitted)

*Figure 4.* Node states and predicted pointers after the last processor step of the DNAR model (with 3 states) trained without hints. Node colors correspond to their states and the green node is the starting node for BFS.

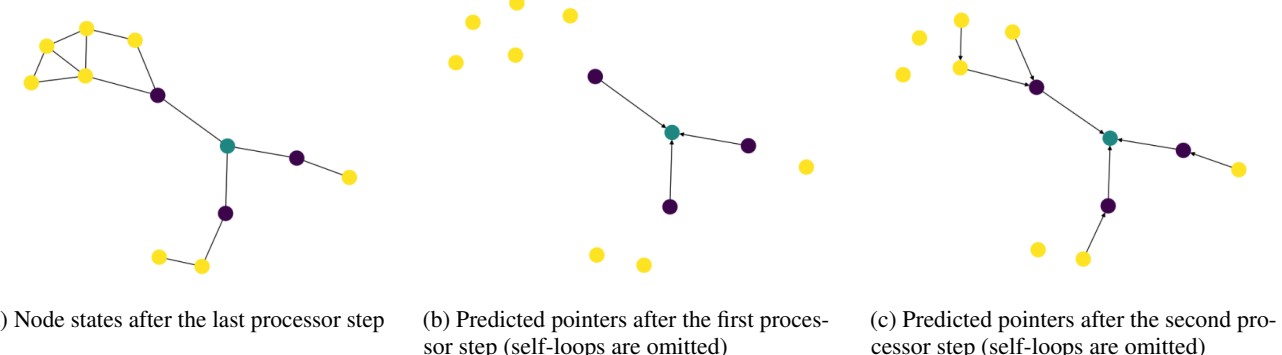

(a) Node states after the last processor step     (b) Predicted pointers after the first proces-    (c) Predicted pointers after the second pro-
                                           sor step (self-loops are omitted)           cessor step (self-loops are omitted)

*Figure 5.* Node states and dynamics of the pointer prediction updates of the DNAR model (with 3 states) trained without hints. Node colors correspond to their states and the green node is the starting node for BFS.

