# OpenReview forum: "Discrete Neural Algorithmic Reasoning"
_ICML.cc/2025/Conference — ICML 2025 poster_

### Official Review · Reviewer_XN76 · 2025-03-09

**Overall Recommendation:** 3

**Summary:**

This paper addresses the problem of neural algorithmic reasoning, where the objective is to train a neural network to mimic each step of a given classical algorithm. The authors propose a novel architecture for this task. They partition the input graph instance into discrete and continuous components and process them separately. Specifically, the discrete component is handled similarly to previous approaches, while the continuous component is used exclusively in the attention weights of the graph neural network. Empirical evaluations demonstrate that their architecture achieves SOTA performance on several algorithmic tasks.

## update after rebuttal
I appreciate the authors' rebuttal and will keep my original score.

**Claims And Evidence:**

Yes, the paper provides experiments and ablation studies to support their claims.

**Essential References Not Discussed:**

No

**Experimental Designs Or Analyses:**

Yes, the experiment is sound.

**Methods And Evaluation Criteria:**

The proposed method seems reasonable (though I still have some uncertainties about the details; see below). In certain algorithmic tasks, it does make sense to handle scalar information differently. The method was tested on a public dataset, but the authors only presented results for a subset of algorithmic tasks, without specifying the evaluation metric. It would be better to clarify the specific evaluation metric used and provide results for all algorithmic tasks in the dataset.

**Other Comments Or Suggestions:**

.

**Other Strengths And Weaknesses:**

- A more comprehensive architecture diagram, including the encoder and decoder modules, should be provided. The current architecture is somewhat confusing (see the questions below).

- The performance on other algorithmic tasks in the benchmark dataset should be presented and the evaluation metric should be clarified.

**Questions For Authors:**

- In an algorithmic task with hints, does the proposed method require invoking the encoder and decoder at each step? As far as I know, previous approaches call the encoder and decoder at every algorithmic step. This actually raises another question I am confused: what exactly does the discrete state refer to? Could you provide a concrete example using BFS? If these discrete states are simply the hints provided in the dataset, then except for the part of handling scalar information, the overall framework doesn’t seem significantly different from previous methods, as prior approaches also produced discrete states via the decoder, which were then fed into the next algorithmic step.

- What are the specific node-level and graph-level evaluation metrics? Could you provide a concrete example?

**Relation To Broader Scientific Literature:**

The paper contributes to the field of neural algorithmic reasoning. The proposed method makes sense and could influence future research in this area, I think.

**Theoretical Claims:**

NA

---

> ### Author Rebuttal · Authors · 2025-04-01
>
> We thank the reviewer for the constructive review and positive feedback! We address the questions below.
>
> > The performance on other algorithmic tasks in the benchmark dataset should be presented
>
> Let us highlight the main part of our contribution: we consider the proposed model in its current form as a potential answer on where perfect generalization might come from. In this sense, we investigate different architectural choices which are needed for different forms of the generalization.
>
> We agree that a more general architecture is of great interest for future work. In its current form, the proposed model is not capable of executing some algorithms from the CLRS-30 benchmark, however, simple modifications can enhance its expressivity, e.g., by supporting more complex aggregations with top-k attention (with fixed k). Also, as we mention in Section 4.1, the model can be extended with additional architectural modifications, such as edge-based reasoning. For example, edge-based attention with top-2 attention (where each edge chooses the top 2 adjacent edges to receive the message from) can implement the sort of triplet reasoning. In our work, we aim to describe the key principles that help to build perfectly generalizable models and do not focus on a general architecture capable of executing a wide range of algorithms.
>
> > what exactly does the discrete state refer to? Could you provide a concrete example using BFS? If these discrete states are simply the hints provided in the dataset, then except for the part of handling scalar information, the overall framework doesn’t seem significantly different from previous methods, as prior approaches also produced discrete states via the decoder, which were then fed into the next algorithmic step.
>
> Yes, except for the part of handling the scalar information the overall framework is similar to prior work. The main remaining difference is that we use discrete bottleneck (essentially feed hint predictions to the processor at the next step) and do not use the hidden states from the previous steps. Most of the previous methods use both re-encoded hint predictions and previous hiddens.
> Our architectural choice is close to ForgetNet (not G-ForgetNet) from Bohde et al. (2024). From this perspective, the difference of our method from ForgetNet can be described as adding discretization to scalar updates, using hard attention, and stepwise training (i.e. training with teacher-forcing). We will add an architecture diagram with additional clarifications to the main text.
>
> > In an algorithmic task with hints, does the proposed method require invoking the encoder and decoder at each step? As far as I know, previous approaches call the encoder and decoder at every algorithmic step.
>
> Yes, similar to prior work, the encoder is the module which maps the discrete state to the high-dimensional vector and the decoder is the module which projects the high-dimensional vectors to the logits of hints/states. Similar to hints reencoding after each step in the prior work, the encoder and decoder invoked in the discrete bottleneck after each processor step.
>
>
> > What are the specific node-level and graph-level evaluation metrics? Could you provide a concrete example?
>
> For all covered problems, node-level metrics represent the accuracy of predicting correct pointers from each node averaged across all nodes in all test graphs. The exception is the MIS problem, where the node-level metric is the accuracy of predicting the correct binary class (in the MIS, not in the MIS) for each node.
>
> Graph-level metrics represent the accuracy of correctly predicting all node-level outputs in a graph, averaged across all test graphs.
> For example, for the BFS problem, output of the algorithm is presented as a tree, where each node points to its parent in the BFS exploration tree (and the starting node points to itself). Pointer is a specific hint/output type which forces each node to point to exactly one node. Technically this is usually implemented via taking the softmax/argmax over the neighbors.
> We will add the example above to the paper.
>
> We thank the reviewer for thoughtful questions and we are happy to discuss further.

---

### Official Review · Reviewer_6CKL · 2025-03-10

**Overall Recommendation:** 3

**Summary:**

Neural reasoners are robust to the noisy data but struggling with out-of-distribution data. Classic symbolic algorithms have complementary features – they are crisp to noisy inputs, but applicable for any out-of-distribution data.

Authors propose a novel approach that guides neural reasoners to maintain the execution of classic symbolic algorithms, so that they could reason with out-of-distribution data. The execution trajectory of classic symbolic algorithm is interpreted as a combination of finite predefined states. The approach identifies discrete states in continuous data flows, and uses a hard attention mechanism to force neural reasoners to align with discrete states.

**Claims And Evidence:**

Authors claim the proposed method is perfect (the word ‘perfect’ or ‘perfectly’ appears 22 times in the paper), evidenced by experiments on SALSA-CLRS benchmark datasets with 100% accuracy.

**Essential References Not Discussed:**

Authors neglected the following two essential papers.

Sphere Neural Networks for Rational Reasoning. https://arxiv.org/abs/2403.15297
Neural Reasoning for Sure Through Constructing Explainable Models. AAAI 2025

These two papers demonstrate one kind of novel neural networks that achieves symbolic-level syllogistic reasoning, with theoretical proof that this network works for any out-of-distribution input data.

**Experimental Designs Or Analyses:**

Authors conducted and analysed experiments within the statistical machine learning paradigm.

**Methods And Evaluation Criteria:**

The proposed approach is an encode-process-decode procedure. Input data is represented as a graph whose node and edge features are encoded into vector embeddings. The processor is a graph neural network that repeatedly updates the vector embeddings.  Hint supervision trains the processor to follow the execution status of the original algorithm. When the processor finishes, the vector embeddings are fed into a decoder network.

**Other Comments Or Suggestions:**

Authors use supervised learning to train the processor to mimic the execution trajectory of classic algorithms. In my opinion, it is impossible to provide data-independent theoretical proof that the trained processor follows the execution trajectory of classic algorithms.

I would like to suggest authors to weaken the claims (it is not necessary to make such strong theoretical claims here, as your experiments are good enough) and mention the methodological limitation above.

**Other Strengths And Weaknesses:**

Researching how neural networks can mimic symbolic algorithms is not only a nice idea but also of huge scientific and practical value.

But, authors’ experiments do not fully support their claim. A neural network perfectly mimic classic algorithm requires the network to achieve 100% on any out-of-distribution datasets. Using benchmark datasets is far beyond sufficient, even these datasets contain out-of-distribution testing items.

**Questions For Authors:**

1. “We also enforce all node and edge features to be from a fixed finite set, which we call states.” Nodes and features are represented by vectors. You enforce them to be from a fixed finite set. Do you mean the members of the set are symbolic states of the classic algorithm?

2. All input data is represented as a graph, and the processor updates features of nodes and edges. Intuitively, what represents the states of classic algorithms, the configuration of the whole graph, or all features of nodes and edges, or features of some nodes or edges?

3.  The main work of the proposed method is to correctly discretize continuous data flow to align with a classic algorithm. This is not yet the whole story to train a neural reasoner to mimic classic algorithms. Classic computing can be roughly stated as data + algorithm + knowledge. How can your method distinguish data from knowledge?

**Relation To Broader Scientific Literature:**

Yes. This important research is in line with the broader scientific research to improve the interpretability and reliability of neural networks.

**Theoretical Claims:**

Authors claim that they built fully discrete neural reasoners for different algorithmic tasks and demonstrated their ability to perfectly mimic ground-truth algorithm execution. Their experiments achieved perfect test scores (100%) on the multiple algorithmic tasks (with guarantees of correctness on any test data).

---

> ### Author Rebuttal · Authors · 2025-04-01
>
> Thank you for the detailed review! We addressed the questions and concerns below.
>
> > I am not very clear about the detailed process that described from Section 3.3 and 3.4. It would be nice that authors can give a concrete example in the supplementary material.
>
> For example, consider the Dijkstra algorithm. The algorithm uses the edge weight to compute the shortest distance from the starting node to each node, building the tree step-by-step. When adding a new node B to the tree and assigning a node A as a parent in the tree, the algorithm computes the distance from the starting node the the node B by summing the distance to the node A with the weight of the edge AB. To mimic this behavior with our model, we use a scalar updater module. At node level, this module receives as inputs the discrete states of the node and its neighbors and the scalar values stored in corresponding edges. Depending on the discrete states, this module updates the scalar values by one of the supported operations, which is described by the formula (lines 228-230 of the paper). In our example, this module will push the scalar (the distance from the starting node to B) from the node A to the edge AB (edge weight) and sum them. Thus, for edges in the shortest path tree, the scalar value on edges represents the exact shortest distance from the starting node, and for other edges the scalar value will represent the edge weight.
>
> This module allows one to perform any (predefined) operation with scalars needed for the algorithms, but with guarantees to choose the exact operation based on the discrete states, thus being robust to OOD scalar values.
> We will add this example (with an illustration) to Appendix.
>
> Additionally, you can find the Appendix C useful for this context.
>
>
> > Authors neglected the following two essential papers.
>
> Thank you for your suggestion, we will cover works on symbolic reasoning, including the mentioned ones, in the background section.
>
> > But, authors’ experiments do not fully support their claim. …  In my opinion, it is impossible to provide data-independent theoretical proof that the trained processor follows the execution trajectory of classic algorithms.
>
> Our claim is based not only on the empirical results, but also on the design of our model. Discrete and size-independent design allows us to unittest all state transitions to ensure that the model will perform correctly on any test data.
>
> From the theoretical perspective, this is similar to proving that a specific algorithm (e.g. bubble sort or BFS) is working correctly for any input (by working correctly we mean that the outputs of the algorithm satisfy a certain condition), but with a difference that we do not need to prove the correctness of the algorithm, we only need to prove that the model will mimic the execution of the algorithm.  Zooming to the node level, at each step, each node needs to select the neighbor and update its own state based on the current discrete state. Both selection and state transition can be tested independently from any input distribution, because there are only finitely many node/edge states. We refer to Appendix B for details of such proof for the BFS problem.
>
>
> > Nodes and features are represented by vectors. You enforce them to be from a fixed finite set. Do you mean the members of the set are symbolic states of the classic algorithm?
>
> Yes. You can find a specific example in Appendix B.
>
> > All input data is represented as a graph, and the processor updates features of nodes and edges. Intuitively, what represents the states of classic algorithms, the configuration of the whole graph, or all features of nodes and edges, or features of some nodes or edges?
>
> In general, the overall state of the algorithm is represented by all features of all nodes and edges. For example, for the BFS problem, at each step, each node is either visited or not and each edge either represents a pointer in the BFS tree or not. At each step, the processor network updates node/edge features with message passing between adjacent nodes.
>
> > The main work of the proposed method is to correctly discretize continuous data flow to align with a classic algorithm. This is not yet the whole story to train a neural reasoner to mimic classic algorithms. Classic computing can be roughly stated as data + algorithm + knowledge. How can your method distinguish data from knowledge?
>
> Zooming to the node level, each node essentially has a state transition matrix (from the architectural perspective this is an MLP that updates node states depending on the current state and the received message from the selected neighbor). In this level, this update is similar to the finite state machine, so the data is the input states and the knowledge is encapsulated in the learned transition rules (e.g., MLPs).
>
> We hope that our response addresses your concerns and will be happy to answer any additional questions during the discussion period if needed.

---

> > ### Comment · Reviewer_6CKL · 2025-04-01
> >
> > --..allows us to unittest all state transitions to ensure that the model will perform correctly on any test data.
> >
> > It is hard to believe unittest can exhaust any test data. Let us take an example of an arithmetic formula, e.g. (4.1*4.5+2.3)*(3.3+52.2/2.2). This formula can be represented by a tree structure (a graph). Unittest needs to test a unit neural module that mimics addition, a unit neural model that mimics multiplication, and a unit neural model that mimics division.  It is hard for me to believe you can enumerate all the real numbers to test the three neural models. I assume you will develop neural networks to mimic addition, multiplication, and division.
> >
> >
> >     --the knowledge is encapsulated in the learned transition rules (e.g., MLPs).
> >
> > This kind of knowledge (e.g. MLPs) is normally called “procedure knowledge”, different from “declarative knowledge”, such as propositional logic with negations. MLPs can only approximate propositional logical reasoning.

---

> > > ### Author Response · Authors · 2025-04-04
> > >
> > > Thank you for being involved in the discussion!
> > >
> > > To address your concern, let us recall how the ScalarUpdate module is designed. In short, at each step each node/edge in a graph has a discrete state and a scalar value. The ScalarUpdate module can select one of the predefined operations depending only on a discrete state (i.e. which operation to perform, e.g. addition, multiplication, no operation) and apply this operation to scalar values. Thus, to understand whether the module mimics the desired logic (e.g. updates the distances in the Dijkstra’s algorithm) we only need to check whether the different discrete states activate the correct operations, but we do not need to check whether the selected operation is performed correctly for each scalar value.
> > > You can find additional examples of the ScalarUpdate module with various predefined sets of operations in Appendix C, as well as a formula describing the roles of discrete states and scalar values.
> > >
> > > We are happy to engage in further discussions. Unfortunately, we are not able to post more comments. However, if needed, we can edit/extend this comment with additional clarifications.

---

### Official Review · Reviewer_KDUc · 2025-03-13

**Overall Recommendation:** 3

**Summary:**

This paper introduces a novel approach to neural algorithmic reasoning by forcing neural networks to maintain execution trajectories as combinations of finite predefined states.

**Claims And Evidence:**

I think the claims made in this submission are strongly supported by the experimentation. The author demonstrate good generalization scores on both in-distribution and out-of-distribution test data across all evaluated tasks.

**Essential References Not Discussed:**

N/A

**Experimental Designs Or Analyses:**

The author compare against multiple baselines including GIN, PGN, and state-of-the-art models, and they evaluate on diverse algorithmic tasks with different computational requirements. They also test on graphs up to 100× larger than training graphs

**Methods And Evaluation Criteria:**

I believe the proposed methods are appropriate and well-motivated. The authors evaluate on the SALSA-CLRS benchmark which provides a challenging testbed with sparse graphs up to 100× larger than training graphs. Both node-level and graph-level metrics are reported, and comparisons are made against several baseline models.

**Other Comments Or Suggestions:**

My major question is can this method scale up to LLM. If the author can show this in an experiment, I will raise my score

**Other Strengths And Weaknesses:**

1. I find the approach conceptually simple yet highly effective
2. Perfect generalization to graphs 100× larger than training examples is impressive
3. The ability to formally verify correctness on any input is a significant advantage over previous approaches


4. The approach sacrifices some expressivity for perfect generalization, which may limit its applicability to certain problems
5. No discussion of the model size, and can it connect to LLM (scale up?)

**Questions For Authors:**

How does your approach scale to algorithms requiring more complex state spaces? For example, algorithms that need to maintain ordered collections or hierarchical structures?

For problems where the ground truth algorithm is unknown, how might we determine the appropriate number of discrete states needed? Is there a principled way to discover minimal state representations?

**Relation To Broader Scientific Literature:**

I think this work makes some contributions to neural algorithmic reasoning. It builds upon prior work on the CLRS-30 and SALSA-CLRS benchmarks while addressing the fundamental limitation of poor generalization to larger graphs.

**Theoretical Claims:**

no theory

---

> ### Author Rebuttal · Authors · 2025-04-01
>
> Thank you for your review and feedback! We address the raised concerns below.
>
> > The approach sacrifices some expressivity for perfect generalization, which may limit its applicability to certain problems
>
> Let us highlight the main part of our contribution: we consider the proposed model in its current form as a potential answer on where perfect generalization might come from. In this sense, we investigate different architectural choices which are needed for different forms of the generalization.
>
> While there are several ways to improve expressiveness without losing the generalization at all, we also can improve expressiveness by reducing the generalization (as we briefly mention in lines 430-435):
> - Removing hard attention: as we demonstrate in our ablation experiments, using regular attention instead of hard attention yields perfect test scores for the BFS problem, but it is possible to construct adversarial examples with large neighborhood sizes where performance drops. While for more complex attention patterns (besides strictly attending to the single node) the OOD performance might be less robust, the expressivity gain is significant.
> - Removing feature discretization, but updating scalars with discrete operations: as shown by prior work and our ablation experiments, learning precise continuous manipulations is non-trivial and small inaccuracies in such manipulations can significantly affect the overall performance of the NARs. Thus, we can use the proposed separation between the discrete and continuous data flows and do not discretize node/edge features at all (and use discretization in ScalarUpdater). However, for non-attention-based models one needs to come up with how scalars will affect the discrete flow.
>
>
> > No discussion of the model size, and can it connect to LLM (scale up?)
>
> For our experiments we use the model with hidden size 128 and a total 400K parameters, which corresponds to the prior work in the field.
> While there is no direct need to scale up the models on covered tasks, none of the proposed ideas are based on the small model sizes. Importantly, there is some connection between our work and [1]. In short, TransNAR[1] discusses a method to enhance reasoning capabilities of large language models with the task-specific GNN-based NAR model, where any NAR model can be used as an “internal tool”. Thus, replacing the baselines GNN with the proposed DNAR model improves the quality of the tool that the language model can use and the overall performance will be limited only by the correctness of the “tool usage”, and not by the inaccuracies of the tool itself.
> However, there are some difficulties in measuring the direct effect of including the proposed DNAR model in the TransNAR pipeline, as the source code for TransNAR is not yet publicly available.
>
>
> [1]  Bounsi et al. (2024), Transformers meet Neural Algorithmic Reasoners
>
>
>
> > How does your approach scale to algorithms requiring more complex state spaces? For example, algorithms that need to maintain ordered collections or hierarchical structures?
>
> We think that for any specific algorithm it is straightforward to modify the architecture (or only the hints, keeping the architecture the same). However, the main challenge arises with the goal of building the universal architecture for different algorithms and data structures.
>
> In its current form, the proposed model is not capable of executing some algorithms from the CLRS-30 benchmark, however, simple modifications can enhance its expressivity, e.g., by supporting more complex aggregations with top-k attention (with fixed k). Also, as we mention in Section 4.1, the model can be extended with additional architectural modifications, such as edge-based reasoning. For example, edge-based attention with top-2 attention (where each edge chooses the top 2 adjacent edges to receive the message from) can implement the sort of triplet reasoning. In our work, we aim to describe the key principles that help to build perfectly generalizable models and do not focus on a general architecture capable of executing a wide range of algorithms.
>
>
> > For problems where the ground truth algorithm is unknown, how might we determine the appropriate number of discrete states needed? Is there a principled way to discover minimal state representations?
>
> We think that different forms of discrete search are possible techniques for this problem. Additionally, possible techniques can be inspired from the finite automata theory, e.g., constructing the automata from the examples of strings from some regular language. After finding some feasible set of states one can iteratively find equivalent states and merge them. However, we leave a deeper investigation of no-hint discrete neural reasoners for future work.
>
> We hope that our response addresses your concerns and will be happy to answer any additional questions during the discussion period if needed.

---

### Official Review · Reviewer_fqbZ · 2025-03-13

**Overall Recommendation:** 3

**Summary:**

The authors define a learning paradigm where they force a neural reasoner to stay exactly on an execution trajectory as provided by the algorithm they aim to imitate, thus achieve perfect generalization. The architecture allows for verification.
They highlight three crucial architectural choices:
feature discretization, hard attention, discrete and continuous data flow separation

## update after rebuttal: i have updated my score, see last comment.

**Claims And Evidence:**

Clear and convincing evidence. Authors do not overclaim.

**Essential References Not Discussed:**

see above

**Experimental Designs Or Analyses:**

The experimental design and the analyses make sense.

**Methods And Evaluation Criteria:**

The evaluation criteria make sense and the method looks sound.

**Other Comments Or Suggestions:**

In 3.3 (and 3.4): I think a simple example (with an example algorithm like dijkstra + its its inputs illustrated) would help to understand why this treatment of the scalars makes sense. You can put 4.4 in the appendix, that doesn’t really need to be in the main text.

**Other Strengths And Weaknesses:**

Strengths:

I enjoy the perfect and provable accuracy with OOD guarantees due to algorithm verification (interpretability)
It is very appealing that the models can do multiple tasks at once, albeit needing task dependent encoder/decoders.

Weakness:

The method mostly needs supervision from the algorithms they aim to mimic to work well, making this much less useful in practice that it could otherwise be. Their experiments without hints in Sec 6. show that for small problems some advances can be achieved, but they do not seem to be compared to existing work.what do you mean by annealing of the attention weights

I am not an expert in this area and I wonder about the scope of this work. It seems very narrow, esp. given the above limitation.

**Questions For Authors:**

What do you mean by annealing of the attention weights?

**Relation To Broader Scientific Literature:**

I am not well versed in the broader literature to gauge this.

**Theoretical Claims:**

There are no significant theoretical claims.

---

> ### Author Rebuttal · Authors · 2025-04-01
>
> Thank you for your review of our paper! We address your questions and concerns below.
>
> > The method mostly needs supervision from the algorithms they aim to mimic to work well
>
> Let us note that for the current state of the field, learning with hints is an important and unsolved problem. E.g., the large body of research (Section 2.1 of the paper), including the state-of-the-art approaches (Bevilacqua et al., 2023; Bohde et al., 2024) heavily rely on different forms of carefully designed step-by-step hints and are not applicable for no-hint learning without additional modifications.
>
> However, we fully agree that learning without hints is an important and challenging problem for further developments of NAR.
> > Their experiments without hints in Sec 6. show that for small problems some advances can be achieved, but they do not seem to be compared to existing work
>
> We will add no-hint scores for baselines to the paper. As we write in Section 6, we focus only on the BFS algorithm as it is well-aligned with the message-passing framework, has short roll-outs, and can be solved with small states count. Even in this setup, we do not outperform baselines.
>
>
> Additionally, we would like to highlight the important property of the proposed models which supports our motivation to focus only on small problems: the correct state transitions can be learned only from a trivially small size.
>
> For example, for the BFS problem, it is enough to use graphs with only 3 nodes to observe that a not_visited node becomes visited or not depending on the received message. However, the subtask of selecting the parent from the multiple visited neighbors requires at least 4 nodes (where the minimum sufficient example is the complete bipartite graph K(2, 2)).
>
> To demonstrate that, we conducted additional experiments to empirically find the smallest training size for perfect fitting of each covered algorithm. For this experiment we used training with hints, but we consider such examples as additional evidence of the prospects of learning with small graph sizes. We train our models for each problem on ER(n, 0.5) graphs for different n and test the resulting models on the graphs with 160 nodes.
>
> Node level scores on graphs with 160 nodes for different training sizes:
>
>
> |              | 3  | 4   | 5   |
> |--------------|----|-----|-----|
> | BFS          | 41 | 100 | 100 |
> | DFS          | 38 | 100 | 100 |
> | Dijkstra     | 13 | 26  | 100 |
> | MST          | 11 | 14  | 100 |
> | MIS          | 79 | 100 | 100 |
> | Ecc. | 45 | 100 | 100 |
>
> Note that the empirical bound is around 4-5 nodes. We leave a deeper investigation of learning models without hints for future work.
> > What do you mean by annealing of the attention weights?
>
> By annealing of the attention weights we mean the convergence to zero of the maximum of attention weights when the count of neighbors limits to infinity. We refer to Appendix A (hard attention) for a specific graph construction which demonstrates that increasing the neighborhood size of a node breaks the ability of that node to select the most important neighbor, which supports our choice of the hard attention for strong size generalization with guarantees.
> > In 3.3 (and 3.4): I think a simple example (with an example algorithm like dijkstra) would help to understand why this treatment of the scalars makes sense
>
> For example, consider the Dijkstra algorithm. The algorithm uses the edge weight to compute the shortest distance from the starting node to each node, building the tree step-by-step. When adding a new node B to the tree and assigning a node A as a parent in the tree, the algorithm computes the distance from the starting node the the node B by summing the distance to the node A with the weight of the edge AB. To mimic this behavior with our model, we use a scalar updater module. At node level, this module receives as inputs the discrete states of the node and its neighbors and the scalar values stored in corresponding edges. Depending on the discrete states, this module updates the scalar values by one of the supported operations, which is described by the formula (lines 228-230 of the paper). In our example, this module will push the scalar (the distance from the starting node to B) from the node A to the edge AB (edge weight) and sum them. Thus, for edges in the shortest path tree, the scalar value on edges represents the exact shortest distance from the starting node, and for other edges the scalar value will represent the edge weight.
>
> This module allows one to perform any (predefined) operation with scalars needed for the algorithms, but with guarantees to choose the exact operation based on the discrete states, thus being robust to OOD scalar values.
>
> We will add this example (with an illustration) to Appendix. Additionally, you can find the Appendix C useful for this context.
>
> We hope that our response addresses your concerns and will be happy to answer any additional questions during the discussion period if needed.

---

> > ### Comment · Reviewer_fqbZ · 2025-04-04
> >
> > Thank you for the additional experiment.
> >
> > Please make sure to include the dijkstra example (and possibly even a visual illustration) into the camera-ready if this paper gets accepted. it helps.
> >
> > I will update my score to 3, as I think this paper gives some interesting insights.

---

### Decision · Program_Chairs · 2025-05-01

**Decision:**

Accept (poster)

**Comment:**

This paper introduces a new architecture for training neural networks to imitate algorithms. It forces intermediate discrete states, and requires supervising on "hints" (intermediate execution states), but is able to perfectly learn an impressive range of nontrivial algorithms. Reviewers unanimously recommend acceptance. I also like the paper, but I think it's a major limitation that it requires this high level of supervision. The authors are upfront about this limitation, and also give interpretability and verification results. Overall, because of the extra supervision, it's not exactly the result that you would want for a neural algorithmic reasoner, but it's a good step forward.